# IB-GAN: Disentangled Representation Learning with Information Bottleneck GAN

## Abstract

We present a novel architecture of GAN for a disentangled representation learning. The new model architecture is inspired by Information Bottleneck (IB) theory thereby named IB-GAN. IB-GAN objective is similar to that of InfoGAN but has a crucial difference; a capacity regularization for mutual information is adopted, thanks to which the generator of IB-GAN can harness a latent representation in disentangled and interpretable manner. To facilitate the optimization of IB-GAN in practice, a new variational upper-bound is derived. With experiments on CelebA, 3DChairs, and dSprites datasets, we demonstrate that the visual quality of samples generated by IB-GAN is often better than those by $\beta$-VAEs. Moreover, IB-GAN achieves much higher disentanglement metrics score than $\beta$-VAEs or InfoGAN on the dSprites dataset.

## 1 Introduction

Learning good representations for data is one of the essential topics in machine learning community. Although any strict definition for it may not exist, the consensus about the useful properties of good representations has been discussed throughout many studies (Bengio et al., 2013; Lake et al., 2017; Higgins et al., 2018). A disentanglement, one of those useful properties of representation, is often described as a statistical independence or factorization; each independent factor is expected to be semantically well aligned with the human intuition on the data generative factor (*e.g.* a chair-type from azimuth on Chairs dataset (Aubry et al., 2014), or age from azimuth on CelebA dataset (Liu et al., 2015)). The learned representation distilling each important factors of data into a single independent direction is hard to be done but highly valuable for many other downstream tasks (Ridgeway, 2016; Higgins et al., 2017b; 2018).

Many models have been proposed for disentangled representation learning (Hinton et al., 2011; Kingma et al., 2014; Reed et al., 2014; Narayanaswamy et al., 2017; Denton et al., 2017). Despite their impressive results, they either require knowledge of ground-truth generative factors or weak-supervision (*e.g.* domain knowledge or partial labels). In contrast, among many unsupervised approaches (Desjardins et al., 2012; Kingma & Welling, 2013; Rezende et al., 2014; Springenberg, 2015; Dumoulin et al., 2017), yet the two most successful approaches for the independent factor learning are $\beta$-VAE (Higgins et al., 2017a) and InfoGAN (Chen et al., 2016).

Higgins et al. (2017a) demonstrate that encouraging the KL-divergence term of Variational autoencoder (VAE) objective (Kingma & Welling, 2013; Rezende et al., 2014) by multiplying a constant $\beta > 1$ induces a high-quality disentanglement of latent factors. As follow-up research, Burgess et al. (2018) provide a theoretical justification of the disentangling effect of $\beta$-VAE in the context of Information Bottleneck theory (Tishby et al., 1999; Tishby & Zaslavsky, 2015)

Chen et al. (2016) propose another fully unsupervised approach based on Generative Adversarial Network (GAN) (Goodfellow et al., 2014). He achieves the goal by enforcing the generator to learn disentangled representations through increasing the mutual information (MI) between the generated samples and the latent representations. Although InfoGAN can learn to disentangle representations for relatively simple datasets (*e.g.* MNIST, 3D Chairs), it struggles to do so on more complicated datasets such as CelebA. Moreover, the disentangling performance of the learned representations from InfoGAN is known as not good as the performance of the $\beta$-VAE and its variant models (Higgins et al., 2017a; Kim & Mnih, 2018; Chen et al., 2018).

Stimulated by the success of $\beta$-VAE models (Burgess et al., 2018; Higgins et al., 2017a; Kim & Mnih, 2018; Chen et al., 2018; Esmaeili et al., 2018) with the Information Bottleneck theory (Alemi et al., 2017; 2018; Achille & Soatto, 2017) in disentangled representations learning task, we hypothesize that the weakness of InfoGAN in the representation learning may originate from that it can only maximize the mutual information but lacks any constraining mechanisms. In other words, InfoGAN misses the term upper-bounding the mutual information from the perspective of IB theory.

We present a novel unsupervised model named IB-GAN (*Information Bottleneck GAN*) for learning disentangled representations based on IB theory. We propose a new architecture of GANs from IB theory so that the training objective involves an information capacity constraint that InfoGAN lacks but $\beta$-VAE has. We also derive a new variational approximation algorithm to optimize IB-GAN objective in practice. Thanks to the information regularizer, the generator can use the latent representations in a manner that is both more interpretable and disentangled than InfoGAN

The contributions of this work are summarized as follows:

1. IB-GAN is a new GAN-based model for fully unsupervised learning of disentangled representations. To the best of our knowledge, there is no other unsupervised GAN-based model for this sake except the InfoGAN's variants (Higgins et al., 2017a; Kim & Mnih, 2018).

2. Our work is the first attempt to utilize the IB theory into the GAN-based deep generative model. IB-GAN can be seen as an extension to the InfoGAN, supplementing an information constraining regularizer that InfoGAN misses.

3. IB-GAN surpasses state-of-the-art disentanglement scores of (Higgins et al., 2017a; Eastwood & Williams, 2018) on dSprites dataset (Matthey et al., 2017). The quality of generated samples by IB-GAN on 3D Chairs (Aubry et al., 2014) and CelebA (Liu et al., 2015) is also much realistic compared to that of the existing $\beta$-VAE variants of the same task.

## 2 PRELIMINARIES

We remind some backgrounds: IB principle in section 2.1 and the connection between $\beta$-VAE and IB theory (Burgess et al., 2018) in section 2.2. Lastly, InfoGAN (Chen et al., 2016) is briefly reviewed in section 2.3.

### 2.1 INFORMATION BOTTLENECK PRINCIPLE

Let the input variable $X$ and the target variable $Y$ distributed according to some joint data distribution $p(x, y)$. The goal of the IB (Tishby et al., 1999; Alemi et al., 2017; 2018) is to obtain a compressive representation $Z$ from the input variable $X$, while maintaining the predictive information about the target variable $Y$ as much as possible. The objective for the IB is

$$\max_{q_\phi(z|x)} \mathcal{L}_{\text{IB}} = I(Z, Y) - \beta I(Z, X) \tag{1}$$

where $I(\cdot, \cdot)$ denotes MI and $\beta \geq 0$ is a Lagrange multiplier. The goal is to obtain the optimal representation encoder $q_\phi(z|x)$ that balances the trade-off between the maximization and minimization of both MI terms. Hence, the IB objective in Eq.(1) provides a natural means for *good representations* by enforcing the representation $Z$ to ignore irrelevant information from the input and simultaneously to be predictive about the target, which can act as a minimal sufficient statistic of $X$ for predicting $Y$ (Tishby et al., 1999; Alemi et al., 2017; 2018).

A growing body of studies (Alemi et al., 2017; Achille & Soatto, 2018; 2017) supports that the learned representations adapting the IB objective tend to be highly efficient and distilled in terms of its code length (Alemi et al., 2018; Shannon et al., 1951). As a consequence, the learned representation is more generalizable and robust to adversarial attack (Alemi et al., 2017), disentangled (Burgess et al., 2018) and invariant to nuance factors (Achille & Soatto, 2017). Moreover, the IB framework prevents weight over-fitting (Alemi et al., 2018; Achille & Soatto, 2018; Vera et al., 2017), and can be used to visualize high dimensional embedding in a low dimensional latent space (Rabinowitz et al., 2018).

## 2.2 $\beta$-VAE FROM THE IB THEORY

$\beta$-VAE (Higgins et al., 2017a) is one of the state-of-the-art unsupervised disentangled representation learning models. The key idea of $\beta$-VAE is to multiply a constant $\beta \geq 1$ to the KL-divergence term of the original VAE's objective (Kingma & Welling, 2013; Rezende et al., 2014):

$$\max_{p_\theta, q_\phi} \mathcal{L}_{\beta\text{-VAE}} = \mathbb{E}_{p(x)}[\mathbb{E}_{q_\phi(z|x)}[\log p_\theta(x|z)] - \beta\text{KL}(q_\phi(z|x)||p(z))], \tag{2}$$

where the encoder $q_\phi(z|x)$ is generally known as the variational approximation to the intractable $p(z|x)$, $p(z)$ is a prior for the latent representation and $p_\theta(x|z)$ is the decoder in the VAE context.

Recently, a notable connection between $\beta$-VAE and the IB theory has been discovered in (Alemi et al., 2017; 2018). Eq.(2) can be derived from the variational approximation to the IB objective Eq.(1). To clarify this connection, see the variational upper and lower bound of the MI:

$$\mathbb{E}_{p(x)}[\mathbb{E}_{q_\phi(z|x)}[\log p_\theta(x|z)] + H(x) \ \leq \ I_q(Z, X) \ \leq \ \mathbb{E}_{p(x)}[\text{KL}(q_\phi(z|x)||p(z))]. \tag{3}$$

The MI in Eq.(3) subscribed with $q$: $I_q(Z, X) = E_{q_\phi(z|x)p(x)}[q_\phi(z|x)p(x)/q_\phi(z)p(x)]$, is called as the *representational* MI[1]. Given that computing marginal $q_\phi(z)$ is intractable, we can use any prior $p(z)$ to substitute for $q_\phi(z)$, forming the variational upper-bound in Eq.(3)[2]. Likewise, we can use any decoder model $p_\theta(x|z)$ to approximate $q_\phi(x|z) = q_\phi(z|x)p(x)/q_\phi(z)$ of the MI, forming the variational lower-bound in Eq.(3). If the target variable $Y$ in Eq.(1) is replaced with $X$, the task is to reconstruct (auto-encode) data from the representation $Z$. The variational lower-bound of Eq.(1) obtained by leveraging the upper and lower bound of MI in Eq.(3) corresponds to Eq.(2)[3].

The disentanglement-promoting behavior of the $\beta$-VAE based on IB theory is discussed in (Burgess et al., 2018). Constraining the MI (or minimizing KL-divergence in practice) forces the encoder to learn representation containing only strongly relevant information to the data reconstruction, while ignoring other unnecessary (or less-necessary) features. The encoder becomes reluctant to use more channels (or dimensions) of the latent vector to lower the MI constraining cost. Hence, the most distinctive and principle features of data are grouped and aligned along with each independent dimension of the representation space.

## 2.3 INFOGAN: INFORMATION MAXIMIZING GAN

Generative Adversarial Networks (GAN) (Goodfellow et al., 2014) establish a min-max adversarial game between two neural networks, a generator $G$ and a discriminator $D$. The discriminator $D$ aims to distinguish well between real sample $x \sim p(x)$ and synthetic sample created by the $G(z)$ with a random noise $z \sim p(z)$, while the generator $G$ is trained to produce a realistic sample that is indistinguishable from the true sample. The adversarial game is formulated as follow:

$$\min_G \max_D \mathcal{L}_{\text{GAN}}(D, G) = \mathbb{E}_{p(x)}[\log(D(x))] + \mathbb{E}_{p(z)}[\log(1 - D(G(z)))]. \tag{4}$$

Under an optimal discriminator $D^*$, Eq.(4) theoretically involves with the Jensen-Shannon divergence between the synthetic and the true sample distribution: $JS(G(z)||p(x))$. However, Eq.(4) does not have any specific guidance on how $G$ utilizes a mapping from $z$ to $x$. That is, the variation of $z$ in any independent dimension often yields entangled effects on a generated sample $x$.

On the other hand, InfoGAN (Chen et al., 2016) is capable of learning disentangled representations. InfoGAN introduces an additional latent code $c$ and encourages it to describe the semantic features of the data. To do so, the training objective of InfoGAN accommodates a mutual information maximization term between the latent code $c$ and the generated sample $x = G(z, c)$:

$$\max_G \min_D \mathcal{L}_{\text{InfoGAN}}(D, G) = -\mathcal{L}_{\text{GAN}}(D, G) + \lambda I(c, G(z, c)), \tag{5}$$

where $I(\cdot, \cdot)$ denote MI and $\lambda$ is a weight coefficient. To optimize Eq.(5), the variational lower bound of MI is also exploited similar to that of the IM algorithm (Barber & Agakov, 2003).

---

[1]We distinguish it from the *generative* in the next section.

[2]The variational inference relies on the positivity of the KL divergence: $\mathbb{E}_{p(\cdot)}[\log p(\cdot)] \geq \mathbb{E}_{p(\cdot)}[\log q(\cdot)]$ for any variational (or approximating) distribution $q(\cdot)$ (Jordan et al., 1999; Wainwright & Jordan, 2008; Alemi & Fischer, 2018).

[3]A constant data entropy term $H(X) = -\mathbb{E}_{p(x)}[\log p(x)]$ is ignored for brevity.

## 3 Approach

We introduce IB-GAN for disentangled representation learning approach in section 3.1, and propose a practical variational approximation for IB-GAN model in section 3.2. Finally, we discuss some distinctive characteristics of the IB-GAN in-depth in section 3.3.

### 3.1 IB-GAN (Information Bottleneck GAN)

Although InfoGAN (Chen et al., 2016) is a fully unsupervised GAN-based approach for learning disentangled representations, its disentanglement performance is, constantly reported, lower than $\beta$-VAE and its variants (Higgins et al., 2017a; Kim & Mnih, 2018; Chen et al., 2018). we hypothesis the weakness of InfoGAN in independent factor learning may originate from the absence of information constraint or any compression mechanism for the representation.

Hence, our motivation is straightforward; we adopt the IB principle to the objective of InfoGAN, presenting Information Bottleneck GAN (IB-GAN). IB-GAN not only maximizes the MI term as the original InfoGAN does, but also constrains the maximization of MI simultaneously as

$$\max_G \min_D \mathcal{L}_{\text{IB-GAN}}(D, G) = -\lambda \mathcal{L}_{\text{GAN}}(D, G) + I^L(z, G(z)) - \beta I^U(z, G(z)), \quad (6)$$

$$\text{s.t. } I^L(z, G(z)) \leq I_g(z, G(z)) \leq I^U(z, G(z)),$$

where $I^L(\cdot, \cdot)$ and $I^U(\cdot, \cdot)$ denote the lower and upper bound of *generative* MI[4] respectively. The parameters $\lambda$ and $\beta$ are the weight coefficients of the GAN loss and the upper-bound of MI, respectively. More details on these parameters are discussed in section 3.3. One important change[5] in Eq.(6) compared to the InfoGAN objective is regularizing the upper bound of MI with $\beta$, analogously to that of $\beta$-VAE and IB theory.

### 3.2 Optimization of IB-GAN

For the optimization of IB-GAN, we here define the tractable variational lower and upper bound of the MI in Eq.(6) using the similar derivation in (Chen et al., 2016; Agakov & Barber, 2006). For notational consistency, we use $p_\theta(x|z)$ to denote the generator $G(z)$. Then, the variational lower-bound $I^L(z, G(z))$ of the generative MI in Eq.(6) becomes

$$I^L(z, G(z)) = \mathbb{E}_{p_\theta(x|z)p(z)}[\log \frac{q_\phi(z|x)}{p(z)}] \leq I_g(z, G(z)) = \mathbb{E}_{p_\theta(x|z)p(z)}[\log \frac{p_\theta(x|z)p(z)}{p_\theta(x)p(z)}]. \quad (7)$$

Since the generator marginal $p_\theta(x)$ is difficult to calculate, a reconstructor model $q_\phi(z|x)$ is introduced to approximate the quantity $p_\theta(z|x) = p_\theta(x|z)p(z)/p_\theta(x)$ in Eq.(7). The lower-bound holds thanks to positivity of KL-divergence. Intuitively, by improving the reconstruction of an input code $z$ from a generated sample $x = G(z)$, we can maximize the lower-bound of MI between the generator and the code $z$.

In contrast to the lower-bound, obtaining a practical variational upper-bound of the generative MI is not trivial. If we follow the same approach in (Alemi et al., 2017; 2018), the upper-bound $I^U(z, G(z))$ of the generative MI becomes

$$I_g(z, G(z)) = \mathbb{E}_{p_\theta(x|z)p(z)}[\log \frac{p_\theta(x|z)\cancel{p(z)}}{p_\theta(x)\cancel{p(z)}}] \leq I^U(z, G(z)) = \mathbb{E}_{p_\theta(x|z)p(z)} \log[\frac{p_\theta(x|z)}{d(x)}], \quad (8)$$

where $d(x)$ is a variational approximation to the generator marginal $p_\theta(x) = \sum_z p(x|z)p(z)$. However, one critical problem of this approach is, in practice, it is difficult to choose or correctly identify the proper approximation model for $d(x)$.

---

[4]The generative MI is described as $I_g(Z, X) = E_{p_\theta(x|z)p(z)}[p_\theta(x|z)p(z)/p_\theta(x)p(z)]$. This initial formulation of the MI is also exhibited in InfoGAN and IM algorithm (Chen et al., 2016; Barber & Agakov, 2003).

[5]We also omit the distinction between $z$ and $c$ for brevity. Hence, $z$ in Eq.(6) is similar to that of $c$ in Eq.(5).

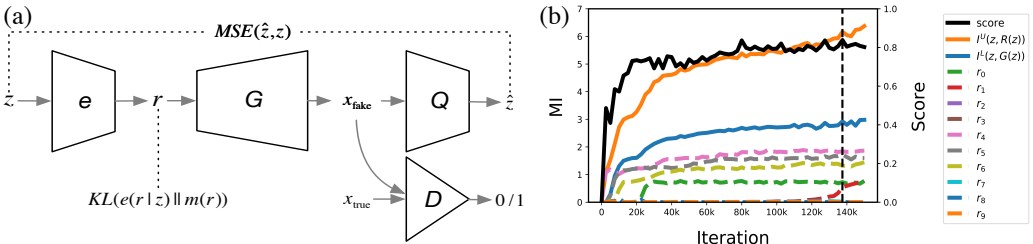

Figure 1: (a) An illustration of IB-GAN architecture. (b) Variations of individual KL-divergence $KL(e(r_i|z)||m(r_i))$ over iterations on dSprites dataset. The sum of these values is the $I^U(z, R(z))$.

---

**Algorithm 1** IB-GAN training algorithm

---

**Input:** batch size $B$, hyperparameters $\lambda$, $\beta$, and learning rates $\eta_\phi, \eta_\theta, \eta_\psi, \eta_w$

    **while** not converged **do**

        Sample $\{z^1, \ldots, z^B\} \sim p(z)$

        Sample $\{x^1, \ldots, x^B\} \sim p(x)$

        Sample $\{r^1, \ldots, r^B\} \sim e_\psi(r|z^i)$ for $i \in \{1 \ldots B\}$

        Sample $\{x_g^1, \ldots, x_g^B\} \sim p_\theta(x|r^i)$ for $i \in \{1 \ldots B\}$

        $g_\phi \leftarrow \nabla_\phi \frac{1}{B} \sum_i \beta KL(e_\psi(r|z^i)||m(r))$

        $g_w \leftarrow -\nabla_w \frac{1}{B} \sum_i \log \sigma(D_w(x_g^i)) + \log(1 - \sigma(D_w(x^i)))$

        $g_\theta \leftarrow \nabla_\theta \frac{1}{B} \sum_i \lambda D_w(x_g^i) - \frac{1}{B} \sum_i \log q_\phi(z^i|x_g^i)$

        $g_\psi \leftarrow \nabla_\psi \frac{1}{B} \sum_i \lambda D_w(x_g^i) - \frac{1}{B} \sum_i \log q_\phi(z^i|x_g^i) + \frac{1}{B} \sum_i \beta KL(e_\psi(r|z^i)||m(r))$

        $\phi \leftarrow \phi - \eta_\phi g_\phi$; $w \leftarrow w - \eta_w g_w$; $\theta \leftarrow \theta - \eta_\theta g_\theta$; $\psi \leftarrow \psi - \eta_\psi g_\psi$

    **end while**

---

In theory, we can choose any model for $d(x)$ (*e.g.* Gaussian), yet any improper choice of $d(x)$ may severely downgrade the quality of synthesized samples from the generator $p_\theta(x|z)$ since the upper-bound $I^U(z, G(z))$ in Eq.(8) is eventually identical to the $KL(p_\theta(x|z)||d(x))$. Moreover, although we express $G(z)$ as $p_\theta(x|z)$ for notional convenience, the probabilistic modeling of generator $G$ will lose the merit of GAN: the likelihood-free (or implicit) modeling assumption.

For this reason, we develop another formulation of the variational upper-bound on the MI term, based on the studies of deep-learning architecture and IB theory (Tishby & Zaslavsky, 2015; Achille & Soatto, 2017; 2018). We define an additional stochastic model $e_\psi(r|z)$ that takes a noise input vector $z$ and produces an intermediate stochastic representation $r$. In other words, we let $x = G(r(z))$ instead of $x = G(z)$, then we can express the generator as $p_\theta(x|z) = \sum_r p_\theta(x|r)e_\psi(r|z)$. Consequently, a practical upper-bound $I^U(z, R(z))$ of the generative MI can be obtained as:

$$I_g(z, G(R(z))) \leq I(z, R(z)) \tag{9}$$

$$I(z, R(z)) = \mathbb{E}_{e_\psi(r|z)p(z)}[\log \frac{e_\psi(r|z)\cancel{p(z)}}{e_\psi(r)\cancel{p(z)}}] \leq I^U(z, R(z)) = \mathbb{E}_{e_\psi(r|z)p(z)} \log[\frac{e_\psi(r|z)}{m(r)}] \tag{10}$$

The first inequality in Eq.(9) holds thanks to the Markov property (Tishby & Zaslavsky, 2015): if any generative process follows $Z \rightarrow R \rightarrow X$, then $I(Z, X) \leq I(Z, R)$. The inequality in Eq.(10) holds from the positivity of KL divergence. Thus, any prior $m(r)$ can be utilized for substituting the marginal $e_\psi(r)$ without affecting the generated samples directly; therefore, this can bypass the difficulty of choosing the prior $d(x)$ in Eq.(8).

Finally, from the variational lower-bound of the MI in Eq.(7) and the newly introduced upper-bound in Eq.(10), the lower-bound of IB-GAN objective in Eq.(6) can be written as:

$$\max_{G, q_\phi, e_\psi} \min_D \tilde{\mathcal{L}}_{\text{IB-GAN}}(D, G, q_\phi, e_\psi) = -\lambda \mathcal{L}_{\text{GAN}}(D, G) \tag{11}$$

$$+ \mathbb{E}_{p(z)}[\mathbb{E}_{p_\theta(x|r)e_\psi(r|z)}[\log q_\phi(z|x)] - \beta \text{KL}(e_\psi(r|z)||m(r))]$$

In other words, the intermediate representation $r$ and the $KL(e_\psi(r|z)||m(r))$ with $\beta$ in Eq.(11) are leveraged to constrain the amount of shared information between the generator $G(z)$ and input $z$. Eq.(11) is optimized by alternatively maximizing the generator $G = p_\theta(x|r)$, the representation encoder $e_\psi(r|z)$, the variational reconstructor $q_\phi(z|x)$ and the discriminator $D$. The IB-GAN architecture is presented in Figure 1(a), and overall training procedure is described in Algorithm 1.

### 3.3 DISCUSSION AND RELATED WORKS

**Connection to rate-distortion theory.** Information Bottleneck theory is a generalization of the rate-distortion theory (Tishby et al., 1999; Alemi et al., 2018; authors, 2019), in which the rate $R$ is the code length per data sample to be transmitted through a noisy channel, and the distortion $D$ represents the approximation error of reconstructing the input from the source code (Alemi et al., 2018; authors, 2019; Shannon et al., 1951). The goal of RD-theory is minimizing $D$ without exceeding a certain level of rate $R$, can be formulated as $\min_{R,D} D + \beta R$, where $\beta \in [0, \infty]$ decides a theoretical achievable optimal frontier in the auto-encoding limit (Alemi et al., 2018).

Likewise, $z$ and $r$ in IB-GAN can be treated as an input and the encoding of the input, respectively. The distortion $D$ is minimized by optimizing the variational reconstructor $q_\phi(z|x(r))$ to predict the input $z$ from its encoding $r$, that is equivalent to maximizing $I^L(z, G(z))$. The minimization of rate $R$ is related minimizing the $KL(e_\psi(r|z)||m(r))$ which measures the in-efficiency (or excess rate) of the representation encoder $e_\psi(r|z)$ in terms of how much it deviates from the prior $m(r)$.

**Disentanglement-promoting behavior.** The disentanglement-promoting behavior of $\beta$-VAE is encouraged by the variational upper-bound of MI term (*i.e.* $KL(q(z|x)||p(z))$). Since $p(z)$ is often a factored Gaussian distribution, the KL-divergence term is decomposed into the form containing a total correlation term (Hoffman & Johnson, 2016; Kim & Mnih, 2018; Chen et al., 2018; Esmaeili et al., 2018; Burgess et al., 2018), which essentially enforces the encoder to output statistically factored representations (Kim & Mnih, 2018; Chen et al., 2018). Nevertheless, in IB-GAN, a noise input $z$ is fed into the representation encoder $e_\psi(r|z)$ instead of the image $x$. Therefore, the disentangling mechanism of IB-GAN must be different from those of $\beta$-VAEs.

From the formulation of the Eq.(11), we could obtain another important insight: the GAN loss in IB-GAN can be seen as the secondary capacity regularizer over the noisy channel since the discriminator of GAN is the JS-divergence (or the reverse KL-divergence) between the generator and the empirical data distribution $p(x)$ in its optimal (Goodfellow et al., 2014; Snderby et al., 2017b). Hence, $\lambda$ controls the information compression level of $z$ in the its encoding $x = G(r(z))$ [6]. In other words, the GAN loss in IB-GAN is a second rate constraint in addition to the first rate constraint $KL(e_\psi(r|z)||m(r))$ in the context of the rate-distortion theorem.

Therefore, we describe the disentanglement-promoting behavior of IB-GAN regarding the rate-distortion theorem. Here, the goal is to deliver the input source $z$ through the noisy channel using the coding $r$ and $x$. We want to use compact encoding schemes for $r$ and $x$. (1) The efficient encoding scheme for $r$ is defined by minimizing $KL(e_\psi(r|z)||m(r))$ with the factored Gaussian prior $m(r)$, which promotes statistical independence of the $r$. (2) The efficient encoding scheme for $x$ is defined by minimizing the divergence between $G(z)$ and the data distribution $p(x)$ via the discriminator; this promote the encoding $x$ to be the realistic image. (3) Maximizing $I^L(z, G(z))$ in IB-GAN indirectly maximize $I(r, G(r))$ since $I(z, G(z)) \leq I(r, G(r))$. In other words, maximizing the lower-bound of MI will increases the statistical dependency between the coding $r$ and $G(r)$, while these encoding need to be efficient in terms of their rate. Therefore, a single independent changes in $r$ must be coordinated with the variations of a independent image factor.

**How to choose hyperparameters.** Although setting any positive values for $\lambda$ and $\beta$ is possible (Alemi et al., 2018), we set $\beta \in [0, 1]$ and fix $\lambda = 1$. We observe that, in the most of the cases, $I^U(r, R(z))$ collapses to 0 when $\beta > 0.75$ in the experiments with dSprites. Although $\lambda$ is another interesting hyperparameter that can control the rate of $x$ (*i.e.* the divergence of the $G(z)$ from $p(x)$), we aims to support the usefulness of IB-GAN in the disentangled representation learning tasks, and thus we focus on the effect of $\beta \in [0, 1.2]$ on the $I^U(r, R(z))$ while fixing $\lambda = 1$. More discussion on the hyperparameter setting will be discussed in Appendix.

## 4 EXPERIMENTS

We experiment our model on various datasets. For quantitative evaluation, we compare methods using the disentanglement metrics proposed in (Kim & Mnih, 2018; Eastwood & Williams, 2018)

---

[6]Recently, authors (2019) provides a theoretical background of viewing the image input as the source channel coding in the framework of deterministic deep learning classification model.

on dSprites dataset (Matthey et al., 2017) (section 4.1). For qualitative evaluation, we visualize latent traversal results on CelebA (Liu et al., 2015) and 3D Chairs (Aubry et al., 2014) (section 4.2).

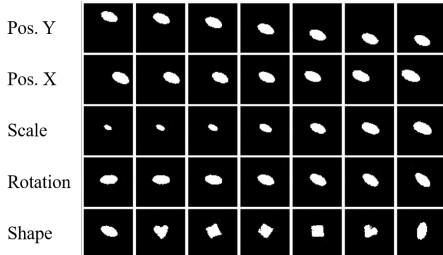

Figure 2: Latent traversals captured by IB-GAN on the dSprites dataset. From top to bottom, each row corresponds to the factors of position Y, X, scale, rotations and shapes.

Table 1: Comparison between methods with the disentanglement metrics in (Kim & Mnih, 2018; Eastwood & Williams, 2018). Our model's scores are obtained from 32 random seeds, with a peak score of (0.91, 0.78). The baseline scores except InfoGAN are referred to (Esmaeili et al., 2018).

| Models | Kim | Eastwood |
|---|---|---|
| VAE($\beta = 1.0$) | $0.63 \pm 0.06$ | $0.30 \pm 0.10$ |
| $\beta$-VAE($\beta = 4.0$) | $0.63 \pm 0.10$ | $0.41 \pm 0.11$ |
| $\beta$-TCVAE($\beta = 4.0$) | $0.62 \pm 0.07$ | $0.29 \pm 0.10$ |
| HFVAE($\beta = 4.0, \gamma = 3.0$) | $0.63 \pm 0.08$ | $0.39 \pm 0.16$ |
| InfoGAN($\lambda = 0.05$) | $0.59 \pm 0.70$ | $0.41 \pm 0.05$ |
| IB-GAN($\beta = 0.14, \lambda = 1$) | $\mathbf{0.80 \pm 0.07}$ | $\mathbf{0.67 \pm 0.07}$ |

We use DCGAN (Radford et al., 2016) with batch normalization (Ioffe & Szegedy, 2015) as our base model for the generator and the discriminator. We let the reconstructor share the same front-end feature with the discriminator for efficient use of parameters as in the InfoGAN (Chen et al., 2016). Also, the MLP-based representation encoder is used before the generator. We train the model using RMSProp (Tieleman & Hinton, 2012) optimizer with momentum of 0.9. The minibatch size is 64 in all experiments. Lastly, we constrain true and synthetic images to be normalized as $[-1, 1]$. Almost identical architectural configurations for the generator, discriminator, reconstructor, and representation encoder are used in all experiments except that the numbers of parameters are changed depending on the datasets. We defer more details on the models and experimental settings to Appendix.

## 4.1 QUANTITATIVE RESULTS ON DSPRITES

Although it is not easy to evaluate the disentanglement of representations, some quantitative metrics (Higgins et al., 2017a; Kim & Mnih, 2018; Chen et al., 2018; Eastwood & Williams, 2018) have been proposed based on the synthetic datasets providing ground-truth generative factors such as dSprites (Matthey et al., 2017) or teapots (Eastwood & Williams, 2018). We verified our approach with the two different metrics (Kim & Mnih, 2018; Eastwood & Williams, 2018) on the dSprites dataset since this setting is tested with many other state-of-the-art baselines in (Esmaeili et al., 2018) including standard VAE (Kingma & Welling, 2013; Rezende et al., 2014), $\beta$-VAE (Higgins et al., 2017a), TC-VAE (Chen et al., 2018) and HFVAE (Esmaeili et al., 2018).

In experiments, we adopt the instance noises technique (Snderby et al., 2017a) since the dSprites images are too simple for the generator of GAN to learn. That is, the intensity distribution of synthetic images is unnaturally narrow (i.e. $[0, 1]$), making the overlapping probability with generated images very low, where the generator may barely learn from the discriminator. Hence, by adding instance noises $\epsilon \sim N(0, \sigma_{instance} * I)$ to both true and generated inputs, we can significantly improve the training stability of GAN models. This may be the reason for the inconsistency between the previous experiments (Kim & Mnih, 2018; Chen et al., 2018) on InfoGAN and our results. For the technical detail, we anneal $\sigma_{instance}$ linearly from 1 to 0 during training for InfoGAN and IB-GAN.

Figure 1(b) shows the variations of $\text{KL}(e(r_i|z)||m(r_i))$ for 10-dimensional $r$ (i.e. $i = 1, \ldots, 10$) over training iterations on dSprites dataset (Matthey et al., 2017). The sum of these values is the upper-bound of MI. We observe that all factors of variations are capped by different values. Similar behavior is exhibited in $\beta$-VAE (Burgess et al., 2018). During training, the encoder $e_\psi(r|z)$ is slowly adapted to capture the independent factors of dSprites dataset as the lower-bound of MI increases. We present the visual inspection of the latent traversal (Higgins et al., 2017a) with the learned IB-GAN model in Figure 2. The IB-GAN successfully learns 5 out of 5 ground truth factors from dSprites dataset, including positions of Y and X, scales, rotations, and shapes, which aligns with the caps on KL scores in Figure 1.(b). More results and discussion about the convergence and the effects of $\beta$ will is in Appendix B.

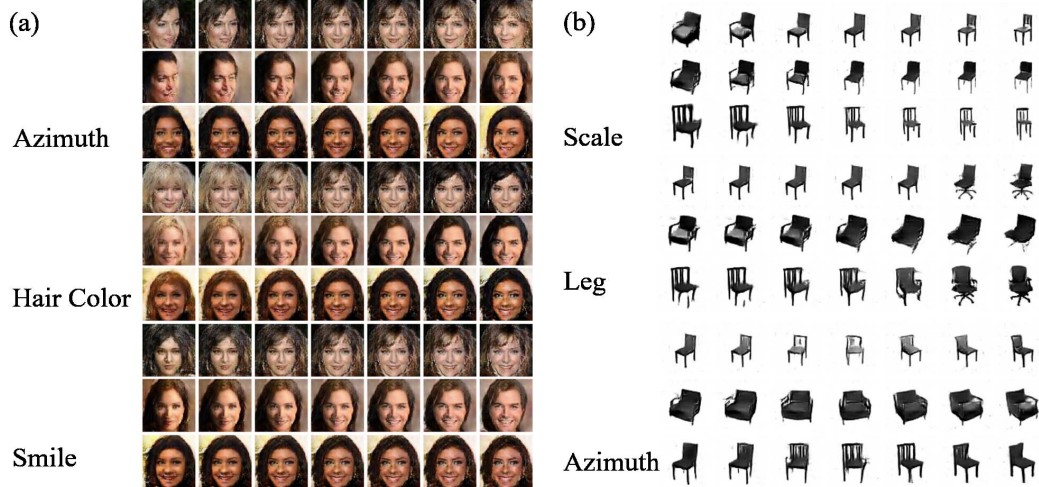

Figure 3: Latent traversals of IB-GAN. IB-GAN (a) learns azimuth, hair color and smile attributes on CelebA, and (b) captures factors of scale, leg and azimuth on 3D Chairs.

Table 1 shows the quantitative results in terms of the two disentanglement metric scores (Kim & Mnih, 2018; Eastwood & Williams, 2018). IB-GAN outperforms other baselines (Kingma & Welling, 2013; Higgins et al., 2017a; Chen et al., 2018; Esmaeili et al., 2018). Interestingly, in our experiments, InfoGAN attains comparable scores to those of other VAE-based models. On the Eastwood's randomforest metric, InfoGAN slightly outperforms other baselines as well, which is consistent with the result of Eastwood & Williams (2018).

### 4.2 QUALITATIVE RESULTS ON CELEBA AND 3D CHAIRS

Following (Chen et al., 2016; Higgins et al., 2017a; Chen et al., 2018; Kim & Mnih, 2018), we evaluate the qualitative results of IB-GAN by inspecting latent traversals. As shown in Figure 3(a), the IB-GAN discovers various human attributes such as azimuth, hair color and smiling face expression. In addition, generated images of the IB-GAN are sharp and realistic than the result of $\beta$-VAE and its variants (Higgins et al., 2017a; Kim & Mnih, 2018; Chen et al., 2018). We also show our qualitative results on 3D Chairs dataset in Figure 3(b). IB-GAN successfully disentangles scales, leg types and azimuth of chairs. These attributes are hardly captured in the original InfoGAN (Chen et al., 2016; Higgins et al., 2017a; Kim & Mnih, 2018; Chen et al., 2018), demonstrating the effectiveness of our approach.

## 5 CONCLUSION

The proposed IB-GAN is a novel unsupervised GAN-based model for learning disentangled representation. We made a crucial modification on the InfoGAN's objective inspired by the IB theory and $\beta$-VAE; specifically, we developed an information capacity constraining term between the generator and the latent representation. We also derived a new variational approximation technique for optimizing IB-GAN. Our experimental results showed that IB-GAN achieved the state-of-the-art performance on disentangled representation learning. The qualitatively generated samples of IB-GAN often had better quality than those of $\beta$-VAE on CelebA and 3D Chairs. IB-GAN attained higher quantitative scores than $\beta$-VAE and InfoGAN with disentanglement metrics on dSprites dataset.

There are many possible directions for future work. First, our model can be naturally extended to adapt a discrete latent representation, as discussed in section 3.3. Second, many extensions of $\beta$-VAE have been actively proposed such as (Burgess et al., 2018; Kim & Mnih, 2018; Chen et al., 2018; Esmaeili et al., 2018), most of which are complementary for the IB-GAN objective. Further exploration toward this direction could be another interesting next topic.

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

# A  MORE DISCUSSION ON IB-GAN

**Reconstruction of input noise z.** The resulting architecture of IB-GAN is partly analogous to that of $\beta$-VAE since both are derived from the IB theory. However, $\beta$-VAE often generates blurry output images due to the large $\beta > 1$ (Kim & Mnih, 2018; Chen et al., 2018; Esmaeili et al., 2018) since setting $\beta > 1$ typically increases the distortion (Alemi et al., 2018). Recently, Alemi et al. (2018) demonstrates the possibility of achieving small distortion with the minimum rate by adopting a complex auto-regressive decoder in $\beta$-VAE and by setting $\beta < 1$. However, their experiment is performed on relatively small dataset (e.g. MNIST, Omniglot).

In contrast, IB-GAN may not suffer from this shortcoming since the generator in IB-GAN learns to generate image by minimizing the rate. Moreover, it does not rely on any probabilistic modeling assumption of the decoder unlike VAEs and can inherit all merits of InfoGANs (*e.g.* producing images of good quality by an implicit decoder, and an adaptation of categorical distribution). One downside of our model would be the introduction of additional capacity control parameter $\lambda$. Although, we fixed $\lambda = 1$ in all of our experiment, which could also affect the convergence or the generalization ability of the generator. Further investigation on this subject could be an interesting future work.

**Behaviors of IB-GAN according to $\beta$.** If $\beta$ is too large such that the KL-divergence term is almost zero, then there would be no difference between the samples from the representation encoder $e_\psi(r|z)$ and the distortion prior $m(r)$. Then, both representation $r$ and generated data $x$ contain no information about $z$ at all, resulting in that the signal from the reconstructor is meaningless to the generator. In this case, the IB-GAN reduces to a vanilla GAN with an input $r \sim p(r)$.

**Maximization of variational lower-bound.** Maximizing the variational lower-bound of generative MI has been employed in IM algorithm (Agakov & Barber, 2006) and InfoGAN (Chen et al., 2016). Recently, Alemi & Fischer (2018) offer the lower-bound of MI, named GILBO, as a data independent measure for the complexity of the learned representations for trained generative models. They discover the optimal lower-bound of the generative MI correlates well with the common image quality metrics of generative models (*e.g.* INCEPTION Salimans et al. (2016) or FID Heusel et al. (2017)). In this work, we discover a new way of upper-bounding the generative MI based on the causal relationship of deep learning architecture, and show the effectiveness of the upper-bound by measures the disentanglement of learned representation.

**Implementation of IB-GAN.** Since the representation encoder $e_\psi(r|z)$ is stochastic, reparametrization trick (Kingma & Welling, 2013) is needed to backpropagate gradient signals for training the encoder model. The representation $r$ can be embedded along with an extra discrete code $c \sim p(c)$ before getting into the generator (*i.e.* $G(r,c)$), and accordingly the reconstructor network becomes $q(r,c|x)$ to predict the discrete code $c$ as well. In this way, it is straightforward to introduce a discrete representation into IB-GAN, which is not an easy task in $\beta$-VAE based models.

Theoretically, we can choose the any number for the dimension of $r$ and $z$. However, The disentangled representation of IB-GAN is learned via the representation encoder $e_\psi(r|z)$. To obtain the representation $r$ back from the real data $x$, we first sample $z$ using the learned reconstructor $q_\phi(z|x)$, and input it to the representation encoder $e_\psi(r|z)$. Therefore, we typically choose a smaller $r$ dimension than that of $z$. For more details on the architecture of IB-GAN, please refer Appendix.E.

**Related Work**. Many extensions of $\beta$-VAE(Higgins et al., 2017a) have been proposed. Burgess et al. (2018) modify $\beta$-VAE's objective such that the KL term is minimized to a specific target constant $C$ instead of scaling the term using $\beta$. Kim & Mnih (2018) and Chen et al. (2018) demonstrate using the ELBO surgery (Hoffman & Johnson, 2016; Makhzani & Frey, 2017) that minimizing the KL-divergence enforces factorization of the marginal encoder, and thus promotes the independence of learned representation. However, a high value of $\beta$ can decrease the MI term too much, and thus often leads to worse reconstruction fidelity compared to the standard VAE. Hence, they introduce a total correlation (Ver Steeg, 2017) based regularization to overcome the reconstruction and disentanglement trade-off. These approaches could be complementary to IB-GAN, since the objective of IB-GAN also involves with the KL term. This exploration could be an interesting future work.

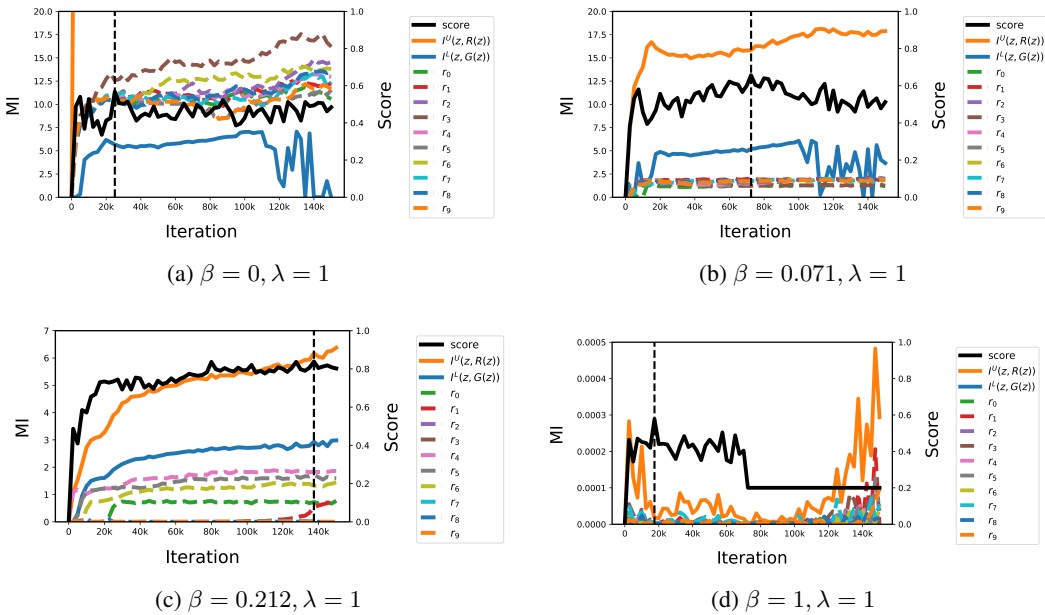

Figure 4: Effects of $\beta$ on the convergence of variational upper-bound (solid orange) and lower-bound (solid blue) of MI with independent KL-Divergences (dashed lines) $\text{KL}(e(r_i|z)||m(r_i))$ for each $r_i$ ($i = 1, \cdots, 10$), and the disentanglement scores Kim & Mnih (2018) (solid black) over 150K training iterations. Vertical dashed black lines represent the iterations at the highest scores.

## B MORE EXPERIMENTS ON DSPRITES DATASET

We investigate the effect of $\beta$ on the convergence of the lower/upper bound of MI in Eq.(1) and the disentanglement metric score (Kim & Mnih, 2018) of the learned representations. We train the IB-GAN model on dSprites dataset (Matthey et al., 2017) while varying $\beta \in [0, 1.2]$ at fixed $\lambda = 1$.

### B.1 EFFECTS OF $\beta$ ON THE CONVERGENCE OF IB-GAN AND DISENTANGLEMENT SCORES

One of the most important hyperparameter in the IB-GAN objective of Eq.(1) is $\beta$ that controls the ratio of the lower-bound $I^L(z, R(z))$ and the upper-bound $I^U(z, R(z))$. Hence, the optimal balance point between the lower and upper bound term is affected by the $\beta$ (Alemi et al., 2018). Each panel in Figure 4 shows the variational lower and upper-bound of MI along with independent $\text{KL}(e(r_i|z)||m(r_i))$ for each $r_i$ ($i = 1, \cdots, 10$) over the 150K training iterations.

As shown in Figure 4a, if $\beta = 0$, the upper-bound of MI in Eq.(11) is ignored and the constraining effect on the representation $r$ disappears. Hence, the lower-bound of MI can quickly increases up to its natural upper-bound[7] similar to the MI lower-bound in InfoGAN. With $\beta = 1$ of Figure 4d, the upper-bound of MI drops down to almost zero and so does the lower-bound. Hence, the representation $r$ is independent of $z$ (i.e. dose not contain any information about $z$) and IB-GAN reduces to vanilla GAN.

When $\beta$ is set properly as in Figure 4c, both lower and upper-bound of MI increase smoothly and the representation encoder $e_\psi(r|z)$ is slowly adapted to capture the distinctive factors of the dataset, where independent KL-divergence $\text{KL}(e(r_i|z)||m(r_i))$ increases one by one by capturing each disentangled attribute. Note that the sum of individual KL scores is the upper-bound of MI $I^U(z, R(z)) = \sum_i \text{KL}(e(r_i|z)||m(r_i))$. We observe that all factors of variations are capped by different values; this behavior is reported as a key element of the disentangled representation learning in $\beta$-VAE (Burgess et al., 2018).

---

[7]The natural upper-bound of MI in InfoGAN depends on the minimum value between $H(z)$ and $H(x)$. But in IB-GAN, typically $dim(r) < dim(z)$, hence $I(z, G(z))$ is usually bounded by either $H(r)$ or $H(x)$

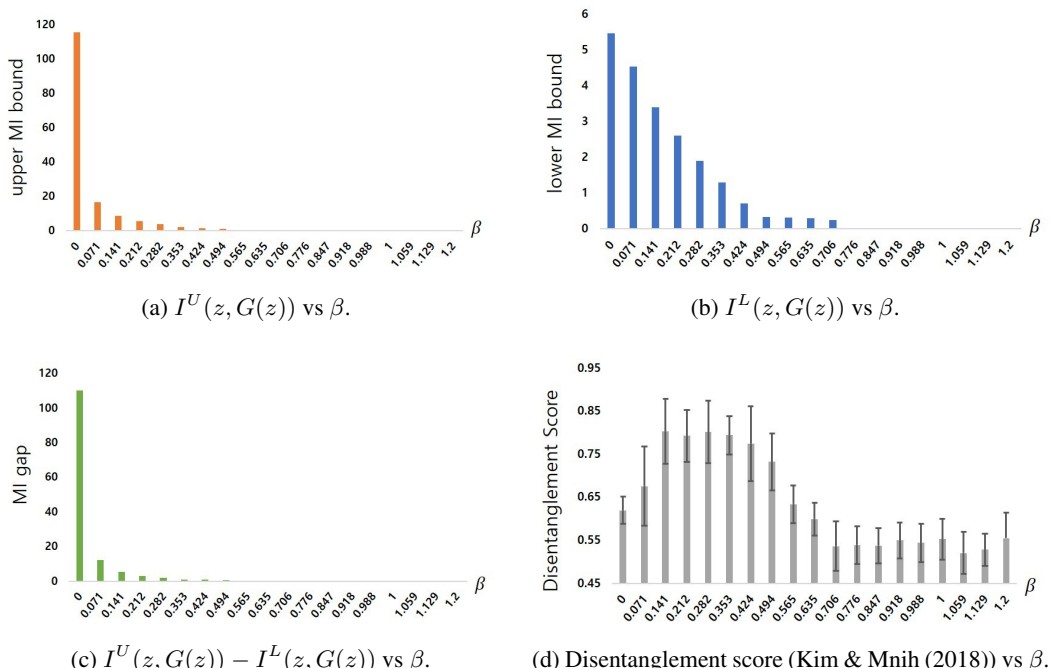

(a) $I^U(z, G(z))$ vs $\beta$.  (b) $I^L(z, G(z))$ vs $\beta$.

(c) $I^U(z, G(z)) - I^L(z, G(z))$ vs $\beta$.  (d) Disentanglement score (Kim & Mnih (2018)) vs $\beta$.

Figure 5: Effects of $\beta$ on the converged upper/lower bound of MI and the disentanglement metric scores Kim & Mnih (2018).

To see the effect of the $\beta$ on the convergence of upper and lower MI bound and the disentanglement score (Kim & Mnih, 2018), we take a median value over the 150k training iterations, and averaged the value over 30 different trials per each $\beta$ in the range of $[0, 1.2]$. Figure 5a and 5b illustrates the expected converged value of upper and lower MI bounds over the different $\beta$. Overall, the upper MI bound tends to decrease exponentially as $\beta$ increases, consequently the lower MI bound decreases as well.

Specifically, $\beta = 0$, the upper-bound MI term disappears in the IB-GAN Eq.(1). Hence, the representation encoding $r$ can diverge from the prior distribution $m(r)$ without any restrictions, resulting in a high value of upper MI bound. Interestingly, the gap between the upper and lower bound is also reduced as the $\beta$ parameter increases as we can see in Figure. 5b. Lastly, Figure.5d shows the effect of $\beta$ on the disentanglement scores. The optimal disentanglement score was achieved when the $\beta$ is around in a range of $[0.1, 0.35]$, and optimal disentanglement score 0.91 is obtained when $\beta = 0.212$ supporting the fact that IB-GAN could control the disentanglement of the learned representation with the upper-bound of generative MI and the varying $\beta$.

## B.2 SAMPLES OF LATENT TRAVERSALS OF THE OPTIMIZED IB-GAN

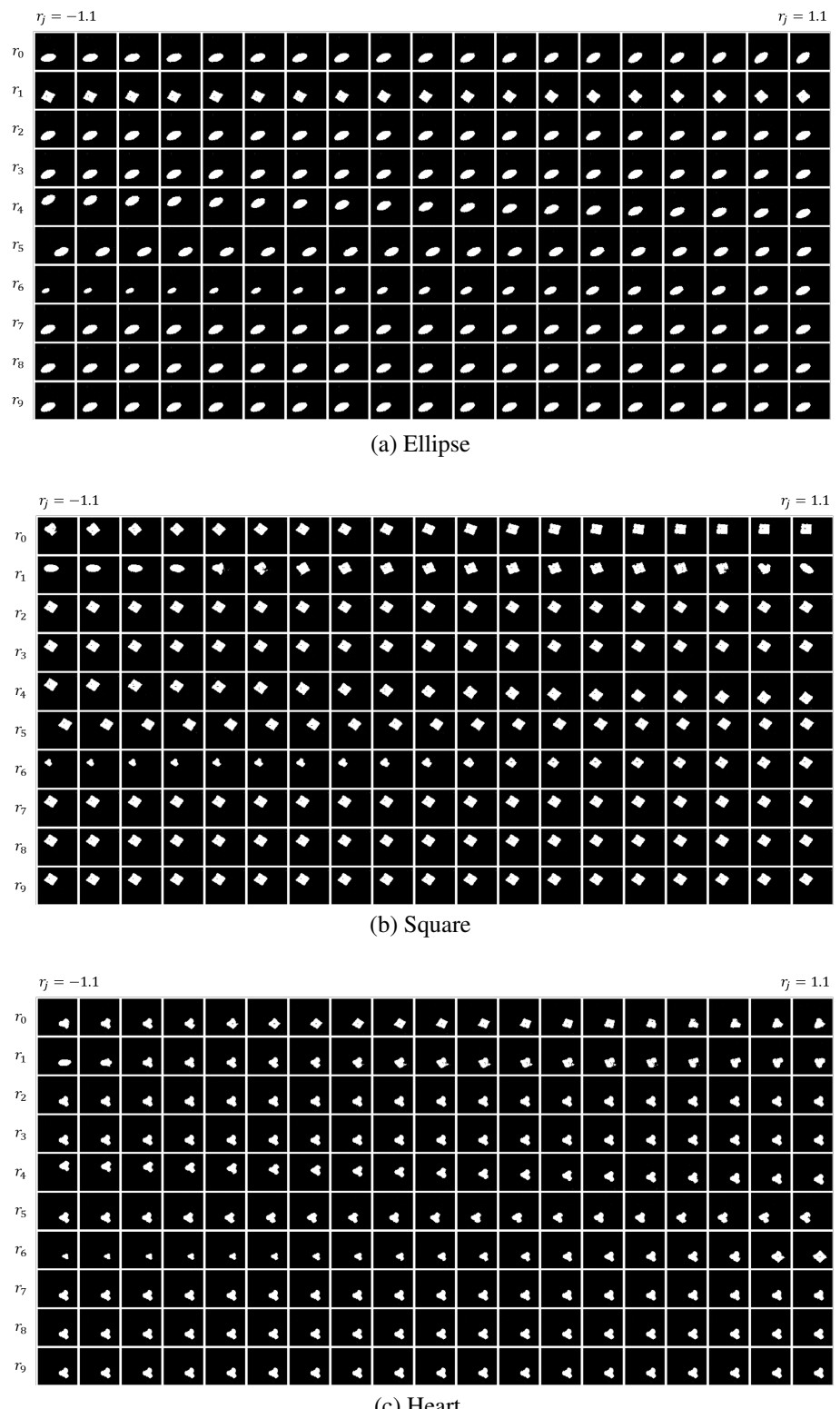

(a) Ellipse

(b) Square

(c) Heart

Figure 6: Some examples of latent traversals of three different base shapes (ellipse, square, and heart) on dSprites dataset with the best parameter setting ($\beta = 0.212$, $\lambda = 1$). IB-GAN successfully captures the five factors of variations: rotations ($r_0$), shapes ($r_1$), positions of $Y$ ($r_4$) and $X$ ($r_5$) and scales ($r_6$). The generator does not reflect the changes in $r_2, r_3, r_7, r_8$ and $r_9$ since they are identical to factored zero mean Gaussian prior $m(r_i)$ and convey no information about $z$. This result align with the Figure.4c: KL-Divergences of these dimensions are nearly zero.

# C  MORE EXPERIMENTS ON CELEBA DATASET

## C.1  SAMPLES OF LATENT TRAVERSALS OF THE OPTIMIZED IB-GAN

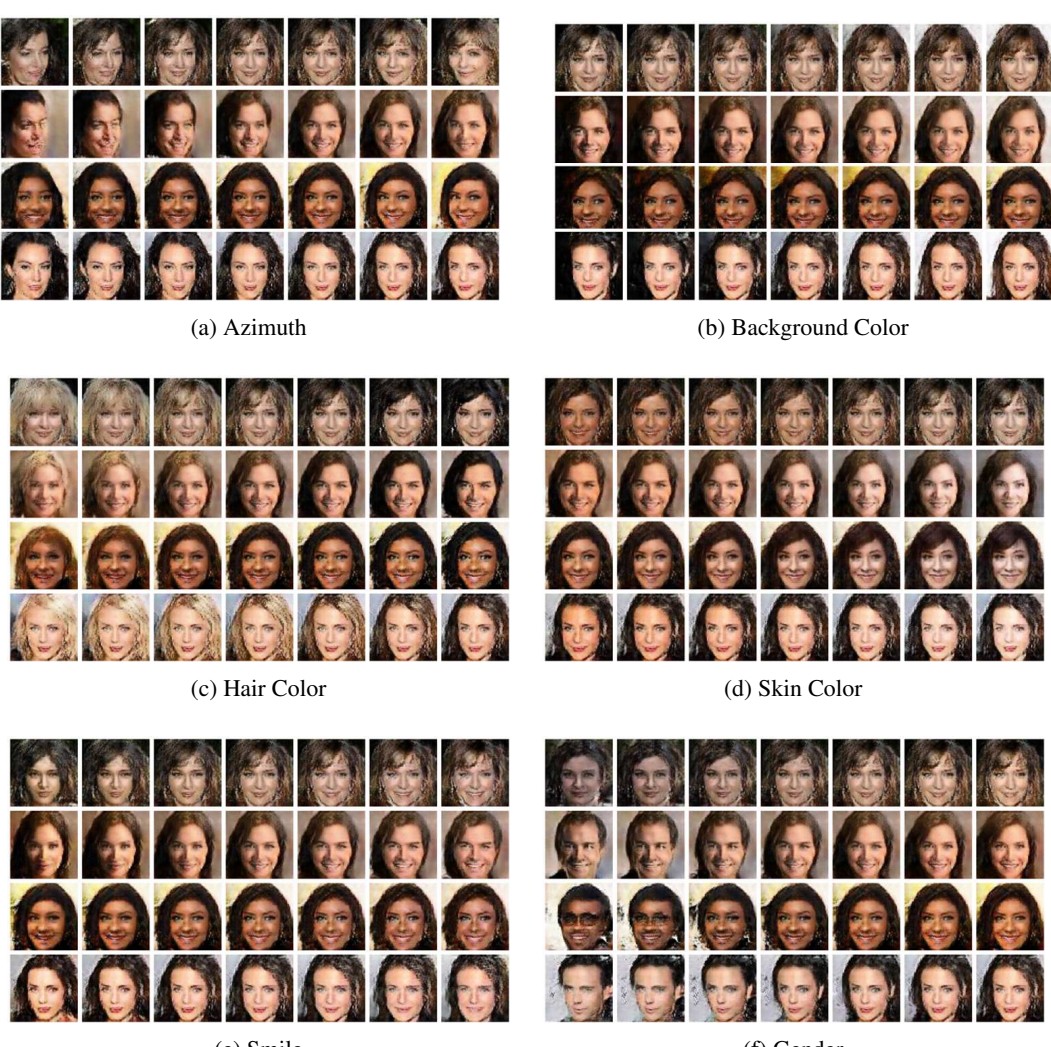

(a) Azimuth

(b) Background Color

(c) Hair Color

(d) Skin Color

(e) Smile

(f) Gender

Figure 7: Latent traversals of four different $r$ vectors on CelebA dataset with the best parameter setting ($\beta = 0.2838, \lambda = 1$).

Following (Chen et al., 2016; Higgins et al., 2017a; Chen et al., 2018; Kim & Mnih, 2018), we evaluate the qualitative results of IB-GAN by inspecting latent traversals. As shown in Figure7, the IB-GAN discovers various human attributes such as (a) azimuth, (b) background color, (c) hair color, (d) skin color, (e) smile, and (f) gender. All of the features in Figure7 and Figure3(a) are captured by one best model with the parameter of $\beta = 0.2838, \lambda = 1$. These attributes are hardly captured in the original InfoGAN (Chen et al., 2016; Higgins et al., 2017a; Kim & Mnih, 2018; Chen et al., 2018), demonstrating the usefulness of the upper-bound of generative MI in IB-GAN. In addition, generated images of the IB-GAN are often sharp and realistic than the results of $\beta$-VAE and its variants (Higgins et al., 2017a; Kim & Mnih, 2018; Chen et al., 2018).

# D   SPECIFICATION OF DATASETS

Table 2: The specification of datasets.

| Dataset | Specification |
|---|---|
| dSprites (Higgins et al., 2017a) | 737,280 binary $64 \times 64$ images of 2D shapes with 5 ground truth factors. Ground truth factors consist of 3 shapes, 6 scales, 40 orientations, and 32 positions of $X$ and $Y$. |
| 3D Chairs (Aubry et al., 2014) | 86,366 gray-scale $64 \times 64$ images of 1,393 chair CAD models with 31 azimuth angles and 2 elevation angles. |
| CelebA (Liu et al., 2015) | 202,599 RGB $64 \times 64 \times 3$ images of celebrity faces consisting of 10,177 identities, 5 landmark locations, and 40 binary attributes. We use the cropped version of the dataset. |

# E  IMPLEMENTATION DETAILS

Table 3 describes the details of hyperparameter settings used in our experiments.

Table 3: The hyperparameter settings for IB-GAN in all experiments.

| Dataset | Optimizer | Hyperparameters | Regularization |
|---------|-----------|-----------------|----------------|
| dSprites | RMSProp(momentum=0.9), LR G/E/Q 5e-5, D 1e-6 iterations 1.5e5 | nc=1, ngf=16, ndf=16, z_dim=64, r_dim=10, $\lambda$=1, $\beta$=0.14 | Instance Noise: $\sigma_{instance}$ is annealed linearly from 1.0 to 0 for 1e5 iterations. |
| 3D Chairs | RMSProp(momentum=0.9), LR G/E/Q 5e-5, D 5e-6 iterations 2e5 | nc=1, ngf=32, ndf=16, z_dim=64, r_dim=10, $\lambda$=1, $\beta$=0.2 | Instance Noise: $\sigma_{instance}$ is annealed linearly from 1.0 to 0 for 1.3e5 iterations. |
| CelebA | RMSProp(momentum=0.9), LR G/E/Q 5e-5, D 5e-7 iterations 2.5e5 | nc=3, ngf=64, ndf=64, z_dim=500, r_dim=15, $\lambda$=1, $\beta$=0.2838 | |

We summarize some implementation details of the models in our experiments on dSprites dataset, 3D Chairs, and CelebA datasets. Table 4 shows the base architectures of IB-GAN for the generator, discriminator, and encoder, while Table 5 shows those of InfoGAN. Table 3 also presents the hyperparameter settings that we use for the models in all experiments.

Table 4: The base architecture for IB-GAN. See Table 3 for hyperparameter setting.

| Generator(G) | Discriminator(D) / Encoder(Q) |
|--------------|-------------------------------|
| Input $z \in \mathbb{R}^{\text{z\_dim}}$ | Input $x \in \mathbb{R}^{64*64*nc}$ |
| FC ngf*2, BN, ReLU | 4x4 conv$\downarrow$, ndf, lReLU |
| FC ngf, BN, ReLU | 4x4 conv$\downarrow$, ndf*2, BN, lReLU |
| FC r_dim*2 $\Rightarrow r_\mu, r_{\log \sigma^2} \in \mathbb{R}^{\text{r\_dim}}$ | 4x4 conv$\downarrow$, ndf*4, BN, lReLU |
| Reparametrization Trick $\Rightarrow r \in \mathbb{R}^{\text{r\_dim}}$ | 3x3 conv, ndf*4, BN, lReLU |
| FC ngf*16, BN, ReLU | 3x3 conv, ndf*4, BN, lReLU |
| FC 8*8*ngf*4, BN, ReLU, | 8x8 conv, ndf*16, BN, lReLU |
| 3x3 conv ngf*4, BN, ReLU | FC z_dim $\Rightarrow z_{recon} \in \mathbb{R}^{\text{z\_dim}}$ |
| 3x3 conv ngf*4, BN, ReLU | FC 1 $\Rightarrow DiscriminatorOutput \in \mathbb{R}^1$ |
| 4x4 conv$\uparrow$ ngf*2, BN, ReLU | |
| 4x4 conv$\uparrow$ ngf, BN, ReLU | |
| 4x4 conv$\uparrow$ nc, Tanh $\Rightarrow x_{fake} \in \mathbb{R}^{64*64*nc}$ | |

Table 5: The base architecture for InfoGAN. All hyperparameters are the same with those of IB-GAN except that z_dim=10.

| Generator(G) | Discriminator(D) / Encoder(Q) |
| --- | --- |
| Input $z \in \mathbb{R}^{z\text{-dim}}$ | Input $x \in \mathbb{R}^{64*64*nc}$ |
| FC ngf*16, BN, ReLU | 4x4 conv↓, ndf, lReLU |
| FC 8*8*ngf*4, BN, ReLU | 4x4 conv↓, ndf*2, BN, lReLU |
| 3x3 conv ngf*4, BN, ReLU | 4x4 conv↓, ndf*4, BN, lReLU |
| 3x3 conv ngf*4, BN, ReLU | 3x3 conv, ndf*4, BN, lReLU |
| 4x4 conv↑ ngf*2, BN, ReLU | 3x3 conv, ndf*4, BN, lReLU |
| 4x4 conv↑ ngf, BN, ReLU | 8x8 conv, ndf*16, BN, lReLU |
| 4x4 conv↑ nc, Tanh $\Rightarrow x_{fake} \in \mathbb{R}^{64*64*nc}$ | FC z_dim $\Rightarrow z_{recon} \in \mathbb{R}^{z\text{-dim}}$ |
| | FC 1 $\Rightarrow DiscriminatorOutput \in \mathbb{R}^1$ |

