# OpenReview forum: "IB-GAN: Disentangled Representation Learning with Information Bottleneck GAN"
_ICLR.cc/2019/Conference_

### Official Review · AnonReviewer1 · 2018-11-06
**Nice idea.  Experiments lacking.**

**Rating:** 4
**Confidence:** 4

**Review:**

Please have your submission proof-read for English style and grammar issues.

This paper introduces the IB-GAN and information bottleneck inspired GAN variant.  The ordinary GAN objective is modified to include a variational lower and upper bound on the generative mutual information.  This should allow one to control the amount of information in the representation of the GAN, in contrast to the InfoGAN which simply maximizes the mutual information.  While lower bounding the generative mutual information is straight forward and only requires a variational inverting network (some q(z|x)) upper bounding the generative mutual information is trickier.  Here the paper offers a very nice solution.  Formally they realize a modified Markov chain   Z -> R -> X where R is made explicitly stochastic.  By Data Processing Inequality I(Z;X) <= I(Z;R) and with a tractable e(r|z), only a variational marginal m(r) is needed to obtain a variational upper bound on the mutual information in the GAN.  This then gives a GAN objective that looks like the information bottleneck interpretation of the VAE.

While the idea for obtaining a variational upper bound on the generative mutual information is novel and clever, the experiments in the paper are lacking.

It should be noted that the variational lower bound on the generative mutual information has already been introduced as the GILBO (generative information lower bound) (arxiv:1802.04874)

I take issue with the discussion in the "Reconstruction of input noise z" section.  It is claimed that beta-VAE "applies the MSE loss to x and uses beta > 1".  VAEs do not have to utilize gaussian observation models and can use powerful autoregressive decoders (e.g. arxiv:1611.02731).

Later down the page it is claimed that when m(r) and p(z) have the same distributional form and dimensionality the R will become independent of Z.  I do not believe this.  What prevents e(r|z) from being a near identity in this situation, for which there could be a large generative mutual information?

The experiments used batch normalization, itself a stochastic procedure that would make their tractable densities incorrect.  There is no discussion of the effect batch norm would have on their bounds.

My principal complaint is the general lack of experimental evidence.  The paper suggests what appears to be a nice framework and simple procedure for controlling the information flow in a GAN.  To do so they introduce two Lagrange multipliers, beta and lambda in their notation (Equation 11) but there are no experiments showing the effect of these two hyperparameters.  They have what should be both an upper and lower bound on the same quantity, the generative mutual information, but these are not shown separately for any of their experiments.  There is no discussion of how tight the bounds are and if they approach each other.  There is no discussion of how the beta and lambda might influence them either individually or jointly.  There is no evidence to demonstrate the effect of constraining the mutual information between X and Z.

In short, the paper offers what appears to be a very clever idea, but does very little to experimentally explore its effects.

---

> ### Author Response · Authors · 2018-11-27
> **To Reviewer1.**
>
> (cont'd)
>
> 4. Batch normalization
> ========================================
> Apparently, we agree that the batch normalization could affect the output of the deep neural network.
> Another hypothesis is that the batch normalization in a deterministic model makes the capacity of the model infinite since it introduces learnable scaling parameters that are not penalized [2].
>
> However, we think the introduction of MI constraint term in our paper gives us way to limit the model capacity (which offsets the effect of batch normalization giving a infinite capacity). In our study, the bottleneck representation encoder e(r|z) is a stochastic model, which means that the deep learning model here is a block box to estimate the mu(z) and var(z) of the distribution of e(r|z) (or the mu(x) and var(x) of the variational reconstructor q(z|x)). Hence, the bias from the batch normalization in IB-GAN architecture converges into the estimation of these parameters. Also, the capacity of both the encoder e(r|z) and the variational reconstructor q(z|x) is bounded by KL(e(r|z)||m(r)) as well. Lastly, other VAE baselines [3,4] in Section 4.1 used batch normalization in their model.
>
> We have conducted a mini test on our model to check if we can see this behavior a bit. We changed the mean of the input Gaussian noise p(z) to some constant C (i.e. 100) while training IB-GAN. It seems that the convergence behavior of IB-GAN objective stumbles a while but very quickly becomes stable, meaning the variational reconstruction q(z|x) starts to predict the input source z well, and overall model capacity was constrained due to the upper-bound. Therefore, we think that our model is not too sensitive to the potential bias effect of batch normalization. However, we believe more investigation on this issue will be very important. We welcome further informations or discussions on this issue.
>
> 5. The tightness of upper and lower bound
> ========================================
> We have updated Figure 4. in Appendix B. to show the changes of the upper-bound and the lower-bound of MI over the training iterations. The gap between the upper and lower bound of MI over the different beta is summarized in Figure 5.(c). Both the upper and lower-bound of MI tend to decrease as the beta increases, which indicates that the upper-bound with beta can constrain the lower-bound well. The gap between the lower-bound and upper-bound also decreases as the beta increases.
>
> 6. The effect of lambda and beta
> ========================================
> The effects of various betas on the variational lower/upper bounds of IB-GAN objective and the disentanglement score are presented in Figure 5. When beta = 0.212, IB-GAN achieved the best disentanglement score of 0.91 in our experiments.
>
> 8. The effect of constraining the MI between X and Z?
> ========================================
> In IB-GAN The generator can learn disentangled representation by constraining the MI between X and Z.  The effect of beta on the disentanglement representation learning is updated in Section 3.3.
>
> We have also tried to describe the disentangling promoting behavior of IB-GAN in terms of the rate-distortion theory [1]. To summarize it, maximizing the lower bound I(z, G(r(z))) in IB-GAN increases I(r, G(r)) since I(z, G(r(z))) <= I(r, G(r)) due to its causal relationship: z->r->x->z’. Maximizing I(r, G(r)) promotes the statistical dependency between r and G(r), while r and x have to be efficient coding which minimizes their excess rate. Therefore, KL divergence with factored Gaussian on r promotes the statistical independence of representation encoder, and statistically distinctive factors or features among the image distribution p(x) must be coordinated with the independent factor of r to maximize the statistical dependency between the source codings: I(z, G(r(z))). For more explanation, please refer to Section 3.3 of updated paper.
>
> Thank you again for the thoughtful insights and comments.
> [1] A. A. Alemi et al. Fixing a Broken ELBO. arXiv:1711.00464 2017.
> [2] A Rate-Distortion Theory of Adversarial Examples, under-review, 2019
> [3] Chen et al., Isolating Sources of Disentanglement in Variational Autoencoders, NIPS 2018.
> [4] Esmaeili et al., Structured Disentangled Representations, arXiv:1804.02086, 2018

---

> ### Author Response · Authors · 2018-11-27
> **To Reviewer1.**
>
> We are sincerely grateful for Reviewer1’s thoughtful review. Please see blue fonts in the newly uploaded draft to check how our paper is updated.
>
> 1. Comparison with GILBO paper
> ========================================
> The lower-bound of I_g(X,Z) in our paper is described in GILBO paper as a “metric” for the learned capacity of any generative models. Specifically, they discovered the correlation between I_g(X,Z) and FID score, implying the higher variational lower bound of MI could indicate that the learned generator is producing more distinctive samples without mode collapses. Thank you for the helpful reference which deepens the understanding of our work. We have referenced this work as you recommended. Still, the novelty of this work lies in the variational upper-bound on I_g(X,Z) and its effect on disentangled representation learning.
>
> 2. VAEs do not have to utilize Gaussian observation models and can use powerful autoregressive decoders (e.g. arxiv:1611.02731).
> ========================================
> Could you checkout that why “Powerful autoregressive decoders are not good for disentanglement learning?” (https://towardsdatascience.com/with-great-power-comes-poor-latent-codes-representation-learning-in-vaes-pt-2-57403690e92b )
>
> Recently, as you mentioned, Alemi et al. [1] demonstrate the possibility of achieving the minimum rate with small distortion as well by adopting a complex auto-regressive decoder in the beta-VAE model by setting beta < 1. But, their experiment is conducted on a relative simple dataset (e.g. MNIST) and not designed for measuring the disentanglement scores explicitly.
>
> In contrast, we achieves the best disentanglement score as exhibited in Section 4.1. IB-GAN also generates images of high quality without any autoregressive decoder. Hence, we believe the visual quality of the generated image could be improved further if a stronger decoder is adapted.
>
> We would like to emphasize the difference between beta-VAE and IB-GAN. The beta-VAE learns to directly generate images from the code z. Instead, the generator in IB-GAN learns to minimize the rate (or the divergence between G(z) and data distribution p(x)) while minimizing the reconstruction error of noise z from its coding x. Given a fixed beta-VAE model, a large beta could consequence in a large distortion. In IB-GAN, however, the generator can learn to generate images of high quality even though the generator is statistically independent of the representation encoder. More discussion is presented in Section 3.3 and Appendix A.
>
> 3. What prevents e(r|z) from being a near identity in this situation, for which there could be a large generative mutual information?
> ========================================
> Thank you for correction our confusion on this. Regardless of its dimension, as the KL(e(r|z)||m(r)) is close to zero, r contains no information about z [1]. Hence, as beta becomes larger IB-GAN reduces to normal GAN. If beta = 0, there is no constraint on the representation r, then IB-GAN reduces to InfoGAN (although bottleneck architecture remains). In Figure 6. some independent dimensions of r vectors do not affect the changes in the generated image, while their KL independence divergence is closed to zero in Figure 1.(b). More discussion can be found in Section 3.3 and Appendix A in our newly updated paper.

---

### Official Review · AnonReviewer2 · 2018-11-07
**Elegant approach, well presented, more experimental validation of the core intuition would have been nice**

**Rating:** 7
**Confidence:** 3

**Review:**

This work addresses the problem of unsupervised disentangled representation learning, and leverages insights and intuitions about utilizing an information bottleneck (IB) approach to encourage disentangling. In particular, building upon insights of how beta-VAE can be understood (and improved upon) by understanding it in terms of IB, the authors propose to modify GANs to include an IB, so as to leverage similar disentangling benefits. The promise is that this approach could utilise the strengths of the GAN framework over VAEs, such as the often sharper reconstructions and the ease of including discrete latents in addition to continuous ones.

To implement their proposal in practice, the authors introduce a neat trick to control an upper bound for the additional mutual information term that the new approach -- termed IB-GAN -- requires. This adds just one layer of complexity to the GAN setup via adding a stochastic representation model between the latent representation and the generator, and has elegant limiting cases that recover both the standard GAN and the InfoGAN approach.

The paper is clearly presented and the intuitive arguments can be readily followed, even though the resulting loss formulation is a bit tricky to justify without expanding upon the underlying motivation.

The approach is tested on three standard datasets and two different metrics that have previously been used for benchmarking unsupervised disentangling, and the results look convincing enough to demonstrate the improvement over existing GAN approaches.

Still, the experimental section is arguably the weakest part of the paper, as there are now stronger beta-VAE variants as baselines available, so I am taking the numbers for VAE-based methods in the quantitative assessment with a grain of salt. More importantly, though, as the motivation of the work is that introducing an information bottleneck is what creates the success in disentangling, it would have been nice to see this effect more clearly broken out in experiments directly demonstrating the effect of beta and gamma on the degree of disentanglement.

Overall, though, this is an interesting contribution to the rapidly developing subfield of unsupervised disentangling, and I would expect the introduction of IB ideas into GAN setups to drive further advances in representation learning techniques.

===
Update:
I am happy with the clarifications and the changes to the manuscript, and have increased my rating accordingly from 6 to 7.

---

> ### Author Response · Authors · 2018-11-27
> **To Reviewer2.**
>
> We thank Reviewer2 for positive and constructive reviews. Below, we respond to each comment in details. Please see blue fonts in the newly uploaded draft to check how our paper is updated.
>
> 1. The effect of beta and gamma on the degree of disentanglement
> ========================================
> The effects of various betas (i.e. [0,1.2] ) with fixed gamma on the disentanglement score is presented in Figure 5. in Appendix B. When, beta = 0.212, IB-GAN achieved a peak disentanglement score of 0.91.
>
> 2. Why IB creates the success in disentangling?
> ========================================
> We have updated our new founding of why our Generator can learn disentangled representation in Section 3.3. Additionally, the effect of beta on the disentanglement metric scores is presented in Figure 5.(d) in Appendix B. We tried to explain the disentangling promoting behavior of IB-GAN with the concepts from the rate-distortion theory in [2].
>
> To summarize it, maximizing the lower bound I(z, G(r(z))) in IB-GAN increases I(r, G(r)) since I(z, G(r(z))) <= I(r, G(r)) due to its causal relationship: z->r->x->z’. Maximizing I(r, G(r)) promotes the statistical dependency between r and G(r), while r and x have to be efficient coding which minimizes their excess rate. Therefore, KL divergence with factored Gaussian on r promotes the statistical independence of representation encoder, and statistically distinctive factors or features among the image distribution p(x) must be coordinated with the independent factor of r to maximize the statistical dependency between the source codings: I(z, G(r(z))). For more explanation, please refer to Section 3.3 of updated paper.
>
> 3. Comparison with stronger beta-VAE variants
> ========================================
> All the beta-VAE based baselines [3,4,5,6] in Section 4.1 are the state-of-the-art models in disentanglement representation learning.
>
> Recently, Alemi et al. [2] demonstrate the possibility of achieving the minimum rate with small distortion by adopting a complex auto-regressive decoder in the beta-VAE model by setting beta < 1. However, their experiment is conducted on a relatively simple dataset (e.g. MNIST) and not designed for measuring the disentanglement scores explicitly.
>
> In contrast, IB-GAN achieved the best disentanglement score, which is summarized in Table 1 in Section 4.1. Moreover, the quality of generated image samples is visually-pleasing without using any complex auto-regressive decoder in IB-GAN. We believe that if we use any stronger autoregressive decoder in our model, the visual quality could be further improved.
>
> We would like to emphasize the difference between beta-VAE and IB-GAN. The beta-VAE learns to directly generate images from the code z. Instead, the generator in IB-GAN learns to minimize the rate (or the divergence between G(z) and data distribution p(x)) while minimizing the reconstruction error of noise z from its coding x. Given a fixed beta-VAE model, a large beta could consequence in a large distortion. In IB-GAN, however, the generator can learn to generate images of high quality even though the generator is statistically independent of the representation encoder.  More discussion in the context of rate-distortion theory [2] is presented in Section 3.3 and Appendix A.
>
> Thank you again for the thoughtful insights and comments.
> [1] C. P. Burgess et al. Understanding disentangling in β-VAE. NIPSw 2017.
> [2] A. A. Alemi et al. Fixing a Broken ELBO. arXiv:1711.00464 2017.
> [3] Higgins et al., beta-VAE: Learning Basic Visual Concepts with a Constrained Variational Framework, ICLR 2017
> [4] Kim & Mnih, Disentangling by Factorising, ICML 2018
> [5] Chen et al., Isolating Sources of Disentanglement in Variational Autoencoders, NIPS 2018.
> [6] Esmaeili et al., Structured Disentangled Representations, arXiv:1804.02086, 2018

---

### Official Review · AnonReviewer3 · 2018-11-07
**A paper with several interesting ideas; Experimental evaluation could do with extra work**

**Rating:** 7
**Confidence:** 4

**Review:**

(Apologies for this belated review)

Summary

The authors propose a GAN-based approach to learning disentangled representations that combines elements InfoGAN with recent Information-Bottleneck (IB) perspectives on variational auto-encoders. In addition to minimizing the normal GAN loss, the authors propose to maximize a lower bound on the mutual information under the generative model, whilst minimizing an upper bound

	Ig[X,Z] = E_p(X,Z)[log p(X,Z) - log p(X) - log p(Z)]

In order to optimize this objective whilst retaining the likelihood-free property of GANs, the authors propose to define a generative model with an intermediate representation r, which allows them to define a likelihood-free generator x = G(r) whilst defining a parametric distribution p(r,z) = eψ(r|z) p(z). This enables the authors to define model architectures that jointly train an encoder qφ(z | x) and a GAN-style generator using an objective that incorporates inductive biases for learning disentangled representations

Quantitative evaluation is performed on d-Sprites (where metrics for disentanglement are evaluated), and qualitative results are shown for Celeb-A and the Chairs dataset.


Comments

I think this is a paper that presents several interesting ideas. Integrating IB-based ideas into the InfoGAN framework is a useful contribution. Moreover, I think that the way they authors integrate a likelihood-free generative model with an inference model is something of a contribution in its own right. I particularly like the idea of the intermediate representation.

Having done some work in this space, I would say that the results on d-Sprites are quite good. Aside from the numerical scores in the table aside, the latent traversals in Figure 2 show a good degree of disentanglement. There is a reason that many of the recent papers don’t show these traversals; it turns out to quite difficult to disentangle shape from the other variables, and even rotation tends to correlate with some of the other latents in many cases.

That said, I would say that the experiments could do with some additional work. I would like to see some discussion of how tight/loose the upper and lower bounds are (some convergence plots would be helpful in this regard). I would also like to see some experiments that evaluate different choices for λ and β (along with some discussion of how these values were chosen – see below). Finally, could the authors find one or two additional datasets? I generally find it difficult to evaluate results on Celeb-A (other than the qualitative evaluation “the images look sharper than those produced by VAEs”). Even something like MNIST/FMNIST would be OK for purposes of evaluating inclusion of Discrete/Concrete variables and/or extrapolation to unseen combinations of factors (as in the Esmaeli et al. paper).

Overall, I would say that this is a potentially strong paper, but that experimental evaluation does need work. I’d be willing to look at an updated version of the paper and adjust my score accordingly if the authors can provide one.

Questions

- Could the authors comment on why they need to set λ=150, β=1? On a quick read, it is not immediately obvious to me why λ > β implies that we will maximize Ig[X,Z] is this simply because maximizing the lower bound will win out over minimizing the upper bound? When we set λ=β, since this would yield a zero loss when the bounds are tight, is that correct? In this case we presumably not necessarily expect to maximize Ig[X,Z] w.r.t. θ?

- What is perhaps missing from this paper is a discussion of *why* maximizing Ig[X,Z] induces a disentangled representation. One hypothesis could be that be that, for a given number of uncorrelated latent variables, a disentangled generator is simply more efficient in terms of the number of distinct samples X it can construct. In this context, it would be interesting if the the authors could report their Ig[X,Z] bounds. In particular, could they compute

	exp[Ig[X,Z]] / N

Intuitively, this number indicates how many examples the generator can produce relative to the number of training examples N. Is it the case that a more disentangled generator is also capable of producing more distinct samples?


Minor

- This is a bit of a pet peeve of mine: Is it really true that GANs lean a representation? A representation is generally a mapping from data to features. A GAN is a mapping from features to data. The authors in this paper do train an encoder to invert the generative model, which learns representation, and certainly a disentangled GAN arguably is useful for more controllable forms of generation in its own right, it just seems that we should not conflate the two.

- Fix: General consent -> General consensus
- Fix: good representation -> good representations
- Fix: such disentangled representation -> disentangled representations
- Fix: (?Higgins et al., 2017b; 2018)

---

> ### Author Response · Authors · 2018-11-27
> **To Reviewer3.**
>
> (cont'd)
>
> 6. Why maximizing I(X,Z) includes a disentangled representation?
> ========================================
> We have updated our new founding of why our Generator can learn disentangled representation in Section 3.3. Additionally, the effect of beta on the disentanglement metric scores is presented in Figure 5.(d) in Appendix B. We tried to explain the disentangling promoting behavior of IB-GAN with the concepts from the rate-distortion theory in [1].
>
> To summarize it, maximizing the lower bound I(z, G(r(z))) in IB-GAN increases I(r, G(r)) since I(z, G(r(z))) <= I(r, G(r)) due to its causal relationship: z->r->x->z’. Maximizing I(r, G(r)) promotes the statistical dependency between r and G(r), while r and x have to be efficient coding which minimizes their excess rate. Therefore, KL divergence with factored Gaussian on r promotes the statistical independence of representation encoder, and statistically distinctive factors or features among the image distribution p(x) must be coordinated with the independent factor of r to maximize the statistical dependency between the source codings: I(z, G(r(z))). For more explanation, please refer to Section 3.3 of updated paper.
>
> 7. Is it the case that a more disentangled generator is also capable of producing more distinct samples?
> ========================================
> Recently, the lower bound I_g(z, G(z)), named GILBO, is also proposed as a “universal measure” for the learned capacity of any given generative models in the paper [3]. They discovered some correlation between the variational lower-bound and FID score, implying the higher variational lower-bound of generative MI indicates the generator produces more distinctive samples without mode collapses or collisions.
>
> Although we did not use the 'exp' function in the equation you suggested, instead, we collected Ig[x_i, z_i] with 20000 samples and averaged it. We believe the lower bound scores in Figure 5.(b) can be seen as the generalization ability of the generator according to [3]. However, the lower bound score is not linearly correlated with FID in Figure 1.(c) in [3]. In fact, the FID score "can" reach good scores at the proper lower bound.
>
> In our studies, we observed at the lower bound scores with beta around [0.141-0.282] in Figure 5, the good disentanglement metric scores are achieved. Similar to GILBO, we could not find a linear relationship between the disentanglement and generalization ability of Generator. Our hypothesis is if we compress the generator too much with the large beta to get disentangled effect, the generalization ability decreases. Therefore, we believe an optimal balance point exists for both good disentanglement and generalization ability. And, we can assume that maximum disentanglement performance a model can achieve depends on its architecture (or its capacity).
>
> Thank you again for the thoughtful insights and comments.
> [1] A. A. Alemi et al. Fixing a Broken ELBO. arXiv:1711.00464 2017.
> [2] A. A. Alemi et al. Deep Variational Information Bottleneck. ICLR 2017.
> [3] A, A. Alemi et al. GILBO: One Metric to Measure Them All. ICLR  2018.

---

> ### Author Response · Authors · 2018-11-29
> **To Reviewer3.**
>
> We thank Reviewer3 for your encouraging and constructive comments. Please see blue fonts in the newly uploaded draft to check how our paper is updated.
>
> 1. The tightness of the bounds
> ========================================
> We have updated Figure 4. in Appendix B. to show the changes of the upper-bound and the lower-bound of MI over the training iterations. The gap between the upper and lower bound of MI over the different beta is summarized in Figure 5.(c). Both the upper and lower-bound of MI tend to decrease as the beta increases, which demonstrates that the upper-bound (with the beta) can constrain the lower-bound well.
>
> 2. The effect of beta (or gamma) on lower/upper bounds and disentangle metric score
> ========================================
> The effects of various betas on the variational lower/upper bounds of IB-GAN objective and the disentanglement score are presented in Figure 5. When beta = 0.212, IB-GAN achieved the best disentanglement score of 0.91.
>
> 3. Adding one or two dataset
> ========================================
> We believe it would be meaningful if we could present our model on additional MNIST/FMNIST dataset with the categorical distributions. However, due to the time constraints, we have focused on the experiment on dSprites to see the effect of beta (and of the upper-bound). These experiments are reported in the Appendix.
>
> 4. Why lambda=150, beta=1? Why “lambda > beta” implies maximizing I(X,Z)?
> ========================================
> According to the rate-distortion theory, the beta is the Lagrange multiplier of an optimization problem, and the ratio between lambda and beta can determine its theoretical optimality [1]. Therefore, we fixed the lambda to 1 in the newly updated paper (in fact we moved lambda to the left term of Eq.(6). And the effect of beta in the range of [0, 1.2] on dSprites dataset is summarized in Appendix B.
>
> To answer the question of why “lambda > beta implies maximizing I(X,Z)?”
> In Figure 5.(b), if beta >= 0.7 and lambda =1 then the upper bound of MI shrink down closer to the zero. In fact, the theoretical optimality of the IB-GAN objective can be decided by the ratio between lambda and beta (or just by the beta when lambda is fixe to 1) [1].
>
> 5. What if lambda = beta. Is this the same as not using I(X, Z)?
> ========================================
> Yes. If we set “beta = lambda”, IB-GAN’s behaviors are similar to those of the normal GAN (although our model is slightly different due to the introduction of stochastic layers before the generator). The convergence behavior when lambda = beta = 1 is exhibited in Figure 4.(d): both MI bounds are very close to zero.

---

### Comment · Area_Chair1 · 2018-12-12
**Question about the baseline results**

I'm worried that the paper the authors copied their baseline quantitative results from didn't conduct a thorough (or any?) hyperparameter tuning in its experiments.

In their rebuttal, the authors state that "All the beta-VAE based baselines [3,4,5,6] in Section 4.1 are the state-of-the-art models in disentanglement representation learning."  However, the baseline results are all taken from "Structured Disentangled Representations"  [https://arxiv.org/pdf/1804.02086.pdf], which doesn't contain any discussion of hyperparameter tuning.  The baselines are also missing the method of Kim & Mnih.

It seems strange that the follow-ups to the beta-VAE were measured to have mostly identical or worse performance.  It also seems strange that they all use identical values of beta, when for instance the beta-TCVAE is weighting a different term and Chen et al. reported optimal beta values that were an order of magnitude different from the beta-VAE.

---

> ### Author Response · Authors · 2018-12-13
> **Dear AreaChair1**
>
>
> We understand your concern that the hyperparameters for the baselines are not thoroughly explored in [1]. Per your suggestion, we will modify the final draft as follows.
> (1) We will update the baseline scores in Table 1 with the scores in the original papers [2,3,4], including Kim & Mnih’s model. We believe these values are obtained based on their optimal settings on the d-sprite dataset.
> (2) Due to delicacy of hyperparameter setting, we will tone down to “our method is comparable” instead of “our method is better than those models.”
> (3) We will make public the source code of IB-GAN for fair comparison in the following research.
>
> Please remind that our model is a new GAN-based model that inherits the merit of GANs but also achieves comparable results to other existing VAE-based models [1,2,3,4]. We also presented a new way of constraining the generative mutual information and discovered an interesting connection between GAN and Information Bottleneck (or rate distortion) theory.
>
> For your information, we quickly summarize the best scores (i.e. disentanglement metric values of [2]) from the original papers on the dSprites dataset.
>
>                      │                [1]                            [2]                           [3]                         [4]
> ════════╪═══════════════════════════════════════════════
>  VAE              │ 0.63(0.06), beta=1   -                                  -                               -
>  Beta-VAE    │ 0.63(0.10), beta=4   0.72*, beta=4            0.78*, beta=2        0.695*, beta=16
>  FactorVAE  │ -                                  0.83*, gamma=35   0.73*, gamma=5   0.79*, gamma=30
>  Beta-TCVAE│ 0.62(0.07), beta=4    -                                  -                               0.78*, beta=4
>  HFVAE        │ 0.63(0.08), beta=4    -                                  -                                -
>  CHyVAE      │ -                                  -                                  0.77*, v=200           -
>                      │ Table 3                       Figure 4                    Figure 4                   Figure 8
> (* denotes the best average scores deduced from the figures in the original papers, where the best average scores are not literally exhibited.)
>
> Our model achieves 0.80 ± 0.07 on average with 32 trials at beta=0.14. According to the table, our IB-GAN shows better scores than most of the reported VAE methods except Kim & Mnih’s model [2]: 0.83. Thus, we can conclude that our IB-GAN is comparable with the state-of-the-art models and the small standard deviation (0.07) implicates that our training is stable.
>
> Thank you for great comments!
> From authors.
>
> [1] Esmaeili et al., Structured Disentangled Representations, arXiv:1804.02086, 2018
> [2] Kim & Mnih, Disentangling by Factorising, ICML, 2018
> [3] Ansari & Soh, Hyperprior Induced Unsupervised Disentanglement of Latent Representations, arXiv:1809.04497, 2018
> [4] Locatello et al., Challenging Common Assumptions in the Unsupervised Learning of Disentangled Representations, arXiv:1811.12359, 2018

---

### Meta-Review · Area_Chair1 · 2018-12-14
**Nice idea, experiments lacking**

**Confidence:** 2
**Recommendation:** Reject

**Metareview:**

Strengths:  This paper introduces a clever construction to build a more principled disentanglement objective for GANs than the InfoGAN.  The paper is relatively clearly written.  This method provides the possibility of combining the merits of GANs with the useful information-theoretic quantities that can be used to regularize VAEs.

Weaknesses:  The quantitative experiments are based entirely around the toy dSprites dataset, on which they perform comparably to other methods.  Additionally, the qualitative results look pretty bad (in my subjective opinion).  They may still be better than a naive VAE, but the authors could have demonstrated the ability of their model by comparing their models against other models both qualitatively and quantitatively on problems hard enough to make the VAEs fail.

Points of contention:  The quantitative baselines are taken from another paper which did zero hyperparameter search.  However the authors provided an updated results table based on numbers from other papers in a comment.

Consensus:  Everyone agreed that the idea was good and the experiments were lacking.  Some of the comments about experiments were addressed in the updated version but not all.

---

> ### Author Response · Authors · 2018-12-23
> **Dear ArearChair1**
>
> We appreciate AreaChair1 and every reviewers for pointing out weak point as well as strong point of our paper. We will keep your advice in mind and try to update or change some of the settings in our experiments to make out our paper much valuable. Thank you.